# The Use of a Two-Dimensional Electrical Resistivity Tomography (2D-ERT) as a Technique for Cadmium Determination in Cacao Crop Soils

**Daniel Bravo [1],\* and Javier Benavides-Erazo [2]**

[1] Laboratory of Soil Microbiology and Calorimetry, Corporación Colombiana de Investigación Agropecuaria –AGROSAVIA, Centro de Investigación Tibaitatá—Km 14, Vía Mosquera-Bogotá, Mosquera CO-0571, Colombia

[2] Laboratory of Physics, University of Nariño, Pasto CO-0572, Colombia; j.benavides.e@gmail.com

\* Correspondence: dbravo@agrosavia.co; Tel.: +57-(1)-4227300 (ext. 1413)



**Featured Application: This study provides new insights into the methodologies to pursue cadmium detection in cacao soils during field tests. This is critical as a first step to produce cocoa that conforms to the European Union and *Codex Alimentarius* regulations, regarding permissible Cd content in products derived from cocoa beans. Thus, the use of the 2D-ERT technique represents a diagnostic methodology to recognize the distribution of cadmium in the soil of cacao growing farms, including both topsoil and subsoil boundaries.**

**Abstract:** Cadmium (Cd) is a non-essential heavy metal naturally occurring in the earth's crust or due to anthropogenic activity. The presence of this metal in cacao farm soils represents a significant issue as levels are now regulated in products derived from cacao beans (*Theobroma cacao* L.). Several strategies have been proposed to measure cadmium levels; however, little is known regarding in situ non-destructive and time efficient techniques to analyze Cd contents in both cacao topsoils and subsoils, particularly nearby the root system. Therefore, this research aims to integrate the physical property of soil resistivity to Cd content in cacao soils. Cd hot spots are estimated from resistivity measurements using a two-dimensional electrical resistivity tomography (2D-ERT) technique and correlated to Cd determination using inductively coupled plasma optical emission spectrometry (ICP-OES). To assess the dynamics of soil Cd content the correlation is discussed with other physical chemical parameters of soils (pH, organic matter, Ca, Fe, and P). The study was performed in 27 cacao farms in Colombia. A farm in Santander district proved to have the highest level of Cd using the correlated techniques (2.76 mg·kg$^{-1}$ Cd and 1815 Ohm·m) followed by farms in Boyacá and Arauca districts (2.6 and 0.66 mg·kg$^{-1}$ Cd, related to 1616 and 743 Ohm·m, respectively). A high correlation between 2D-ERT and Cd determination ($R^2$ = 0.87) was found. The discussion regarding the soil parameters analyzed suggests that the 2D-ERT technique could be used as a preliminary approach to explore Cd distribution in cacao soils.

**Keywords:** cadmium; cacao; two-dimensional electrical resistivity tomography (2D-ERT); ICP-OES; phosphorus (P); soil organic matter (SOM)

---

## 1. Introduction

Cadmium (Cd) presence in cacao soils is an issue of significant concern within certain areas where cacao is farmed in Colombia [1]. Therefore, diagnostic methods based on soil geophysical properties such as the resistivity of soils should be considered to correlate and understand the dynamics of the metal with other physical and chemical parameters of Cd-enriched areas [2]. Hence, the assessment of

parameters such as soil pH, soil organic matter (SOM), soil ionic content of Ca, Fe, and P should be interpreted according to the resistivity responses of Cd-enriched soils.

Since the origin of Cd in cacao crops from South and Central America is due mainly to geogenic conditions [3], it is important to develop strategies to perform sensitive diagnosis of Cd using non-destructive and high-throughput approaches. Furthermore, because the distribution of Cd in soils tends to be heterogeneous, assessing the distribution of this metal in subsoil only using spectrometric quantifications is not an easy task. It requires the use of further approaches involving an integrative analysis of the physical, chemical, and micro-biological properties of the cacao farm soil [4–6].

Regarding the complexity of the potential natural sources of Cd in cacao, both biomineralization and bioweathering have been demonstrated as the biogeochemical processes that influence the rates of availability and precipitation of Cd in soil solution, which may define the ratio of assimilated Cd into the cacao trees [1]. Although the specific mechanisms involved in each of the above mentioned processes remains unknown and may vary on the biological aspects of the biogeochemical pathway (i.e., due to the key role of bacterial metabolic capacities to deal with Cd at the subsurface), the interaction of both biological and chemical factors occurs frequently and influences Cd debris throughout the system. Thus, a proxy to study its distribution and dynamics should be inferred through the physical changes of Cd-like compounds present in enriched or contaminated soils.

Some geophysical studies have noted the use of two-dimensional electrical resistivity tomography (2D-ERT) to analyze the composition of rocks or mineral outcroppings related to cadmium in the subsurface of bulk-type soils [7–9]. The use of the 2D-ERT technique can be considered for determining Cd in cacao for at least three reasons: One, it is a non-invasive, non-destructive technique [7], which determines the mineralogical content of Cd-like solid-phase compounds in subsoil. Second, because cacao farm soils are extremely heterogeneous in physical properties, the study implies the integration of resistivity measurements with the intrinsic physical-chemical composition of soil using parameters such as pH, SOM, and ionic Ca/Fe/P ratios. Third, 2D-ERT is a technique that differentiates geological structures based on resistivity profiles and visualizes Cd outcroppings occurring in situ.

The 2D-ERT method uses electrodes distributed along a line. Each electrode is connected to others, allowing an electrical current flow in soil depth [10]. The distribution of the electrical flow occurs due to the disposition of two electrodes at the soil surface. The determination of resistivity at the subsoil surface is obtained using another pair of electrodes. Thus, the magnitude of the measurement primarily depends on the distribution of resistivity in the subsurface soil structure, the distance between the paired electrodes supplying the electrical signal and the flow of electrical current [8,11]. The theory behind resistivity measurements related to the use of Ohm's law that is determined by an electrical signal of current. Applying the Wenner method of electrode configuration it is possible to map the resistivity at each discharge with high sensitivity and accuracy [12], and it is possible to preselect zones at the subsurface with resistivity related to clay content and the underlying geological parent material [13]. This geophysical method is useful to analyze heavy metal distribution in soils; however, to our knowledge, there are no studies that use the 2D-ERT technique to analyze the Cd distribution in the rhizosphere of cacao farm soils.

Therefore, this study is the first study using the technique 2D-ERT to assess Cd distribution in cacao soils. It provides evidence that the 2D-ERT technique is a useful tool to understand Cd distribution and provides an accurate overview of the issue at the farm level. It is also suggested that 2D-ERT profiles should be made routinely before taking soil samples and quantifying Cd in cacao farms.

## 2. Materials and Methods

### 2.1. Calibration of 2D-ERT In Vitro

The calibration was performed in the Laboratory of Physics at the University of Nariño, in Pasto, Colombia. The resistivity measurements of cocoa farm soil samples were adjusted to small scale

processes [14]. Before calibration the first measurements were done in soil samples with no Cd amendment. The aim of performing a calibration was to correlate the resistivity values with the Cd content of soil samples amended with pure reagents Cd. These values would be expected to be found in Cd enriched or contaminated soils. Two controls of Cd sources were used. The first soluble source was cadmium sulphate powder ($\geq$99.99% *w/w* purity, Sigma-Aldrich, St. Louis, MO, USA). The second non-soluble source corresponds to cadmium carbonate powder (otavite $\geq$99.99% *w/w* purity, Sigma-Aldrich, St. Louis, MO, USA). Whereas $CdSO_4$ is soluble in water at 25 °C (76.7 g $\times$ 100 mL $H_2O$), $CdCO_3$ is not [15]. However, HCl can be used to solubilize $CdCO_3$ [16]. Therefore, 2.5 mL of HCl 36% (*w/v*) was added to 1 g $CdCO_3$ 99.99% (*w/w*) and gently mixed for 10 min at room temperature using a vortex. In both cases, the Cd sources were amended to soils collected from non-Cd contaminated cacao farm ($\leq$0.02 mg·kg$^{-1}$ Cd). These farms were located in the same districts as the contaminated farms.

Two hundred and fifty grams of soil samples were mixed for 10 min with four different concentrations of each Cd source. The treatments consisted of an amendment of 0, 1, 2, and 5 g of the Cd source. The mixture was completed by adding 70 mL of nanopure sterile water and depositing the samples in three blocks of 125 cm$^3$ plastic cube molds (triple mold buckets of cement and mortar, ref. INV-E 323, Pinzuar Ltd.a. Bogotá, Colombia) for physical analysis. After drying the samples at room temperature (20 °C $\pm$ 0.2 °C on average) for 96 h, four resistivity testing electrodes (Stanley steel iron based alloy electrodes, ref. AISI 316 L Hilo 5 mm diameter FF215140, Goodfellow SARL, Lille, France) were displaced at an "*a*" distance of 1 cm, each located equidistantly within the cube (Figure 1A), to produce an average measurement of Cd disposition in the sample. Additionally, a "*b*" distance was used between the external electrodes. The electrical resistivity was measured and quantified as the mean values of a Wenner four electrode array, according to the American Section of the International Association for Testing Materials (ASTM) protocol G 57 [17].

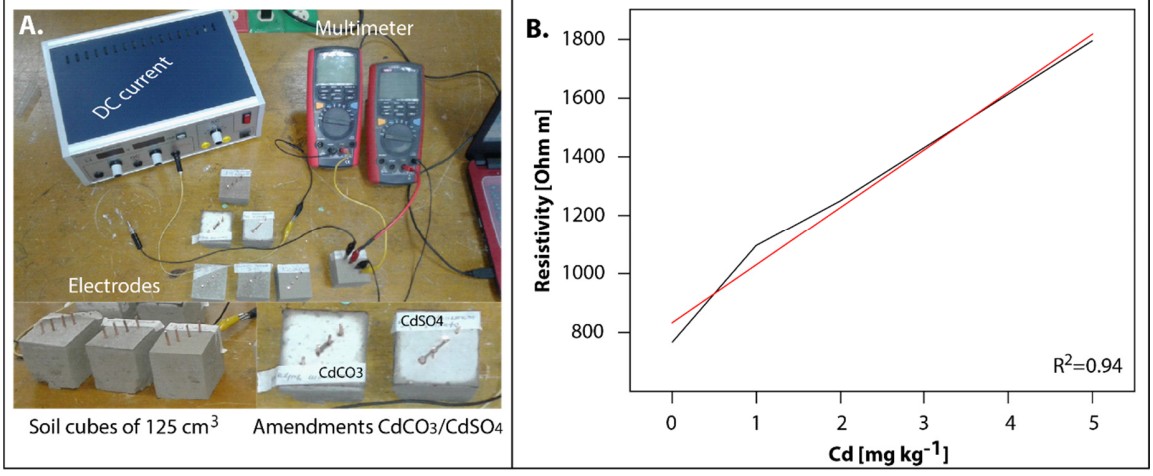

**Figure 1.** Calibration of the two-dimensional electrical resistivity tomography (2D-ERT) technique in vitro. (**A**) The equipment was set for two pure Cd reagents ($CdCO_3$ and $CdSO_4$), measured in four concentrations and in triplicate using a simple electronic device, consisting of the voltage current power, a multimeter, and electrodes; (**B**) the correlation between resistivities and Cd concentrations using cadmium sulphate (red line) and cadmium carbonate (black line). The correlation was high according to the coefficient of determination, as shown in the plot. When no Cd was added the resistivity remained lower than 800 Ohm·m.

The measurement setup was carried out according to a previous study [18]. In summary, the electrodes were connected to a high voltage power supply (SF-9585A: PASCO, Tecnodidácticas Ltd.a. Bogotá, Colombia). The electrodes were also pinned to a digital multimeter (UNI-T UT71D, UNI-T, Hong Kong, China) to register both the current and voltage parameters (see Figure 1A). Fifty replicated measurements were recorded per treatment; therefore, the results were expressed as the mean of the treatments.

A linear regression was made to analyze the correlation between resistivity values with the Cd concentrations of the Cd treatments (see Figure 1B).

## 2.2. Sampling Zones

The study was performed in 27 cacao farms located in three districts in Colombia. The districts were Arauca, Boyacá, and Santander, mainly in the municipalities of Arauquita, Muzo, and San Vicente de Chucurí, respectively (see Figure 2). However, the study also includes other municipalities in Arauca, namely, Saravena and Tame; Boyacá, namely, Maripí and Pauna; and Santander, namely, El Carmen de Chucurí and Rionegro. The farms were selected for cadmium assessment due to previous reports of high levels of cadmium in cacao beans [1]. The plantations were 4 years old and the expected geological depositions and Cd-fluxes occurring at the farms were important criteria for selection. The experimental setup carried out within each farm consisted of three trial pits of 1 $m^3$ of area located in perpendicular lines across the cultured hectare.

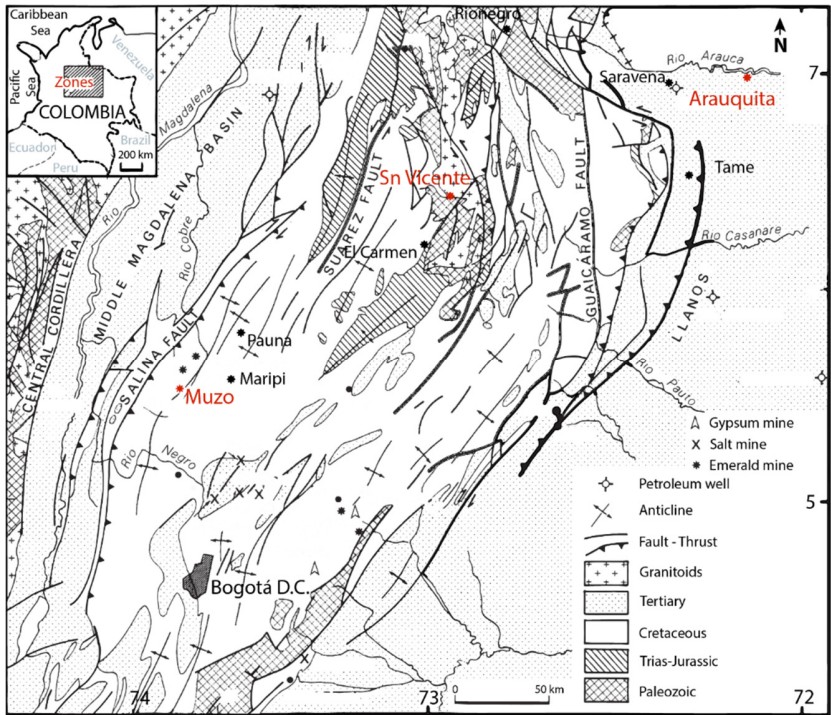

**Figure 2.** Locations of the assessed farms in the municipalities of San Vicente de Chucurí, Arauquita, and Muzo (in red), part of the major cacao producing districts in Colombia (Santander, Arauca, and Boyacá, respectively). As shown on the map, the presence of geological faults/granitoids formations, emerald mines, and petroleum pipelines near the farms are highlighted.

## 2.3. 2D-ERT Measurements in Farms

Three prospecting lines were performed per farm. The Wenner model configuration was selected for this study [8]. Thirty-six electrodes per line were placed at an equally spaced distance of *a* (0.35 m) covering a surface distance *b* (12.25 m) between the first and last electrodes on the line. Soil depth probing was achieved by changing the inter-electrode spacing *a* in successive increasing passes. The prospecting lines were placed in parallel and covered 2 ± 0.1 m of soil depth. The lines were separated from each other by 66 cm on the surface. One transversal line was carried out to fit the resistivity values. A direct electrical current was supplied into the subsoil to generate a difference of voltage measured in surface. The tomography was performed using the software Res2Dinv (GEOTOMO, Penang, Malaysia). The 2D-ERT values were expressed in Ohm·m. Each plot showed in the results section corresponds to the "inverse model resistivity section".

### 2.4. Cd Determination and Physical Parameters

Once the data profiles 2D-ERT obtained in situ were reviewed, one trial-pit was made to collect soil samples per soil boundary to quantify Cd. The soil parameters pH, SOM, P, Ca, and Fe were assessed from the soil samples collected at the trial pits. The Cd levels were then compared with the resistivities obtained from the same spots on the soil profiles. Pseudo-total Cd content was determined using inductively coupled plasma optical emission spectrometry (ICP-OES) (Thermo Scientific ICAP 6500 duo, Waltham, MA, USA) following the previous study [1]. A method recommended by the Environmental Protection Agency (EPA 3050B) was used as the conventional pseudo-total digestion method [19]. The soil samples' pH, SOM, and the ionic content of Ca, Fe, and P were also analyzed. Soil pH was determined with a field pH meter (YSI 556 Handheld Multiparameter Instrument, Yellow Springs, OH, USA) on saturated soil paste (soil to water 1:1), measuring at each boundary of soil surrounding a cacao root in the trial pits. Phosphorous (P) concentration was determined according to the standard Olsen method [17]. The determination of the micronutrients Ca and Fe was also achieved using ICP-OES. The SOM was measured as organic carbon content, following the Walkley Black method [20]. This protocol is based on the reduction of $Cr_2O_7{}^{-2}$ using 100 mg of a composite soil sample, adding 10 mL of 1N $K_2Cr_2O_7$ and gently mixed. Then, 5 mL of 95% $H_2SO_4$ was added slowly over 3 h. After the exothermic reaction, 35 mL of deionized sterile water were added, and the matrix was left to stand overnight. One mL of the supernatant was used for a reading in a spectrophotometer (FastTrack™ UV Vis, Mettler Toledo, OH, USA), at 585 nm wavelength. The measurements were performed in triplicate and data are expressed as the average of the replicates.

### 2.5. Data Analysis

A linear correlation was carried out between the resistivity and soil Cd content, comparing the accuracy of both 2D-ERT and ICP-OES techniques. A further comparison was performed between the vertical distribution of pH, resistivity, and Cd content. The soil parameters of resistivity, pH, SOM, Ca, Fe, and P were also compared to Cd content using a principal component analysis (PCA) to find out the parameter most related to the Cd distribution in samples. The data were plotted as an average of the replicates.

## 3. Results

### 3.1. Standard Curve of Resistivity Under Controlled Conditions

The response of resistivity to increasing concentrations of Cd added from each pure reagent was well correlated and matched the differences between the single Cd sources to the specific ranges of soil resistivity, as shown in Figure 3.

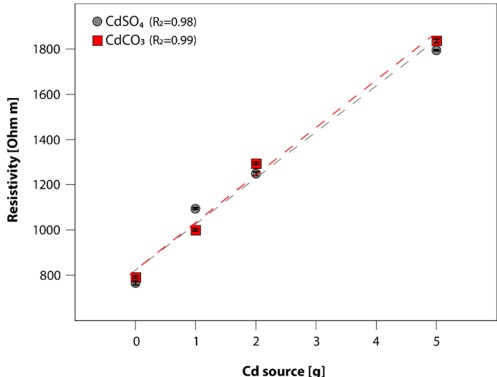

**Figure 3.** Lineal regression between resistivity values [Ohm.m] and four Cd concentrations [g] amended to controlled experiments during the standardization of the 2D–ERT technique into non-naturally contaminated soil samples. SD is shown as vertical bars.

The coefficient of determination was high for each Cd source (either cadmium sulphate or cadmium carbonate) analyzed following a linear regression ($R^2 = 0.94 \pm 0.2$ on average for both Cd sources) as shown in Figures 1B and 3. The 2D-ERT technique was adjusted to separate resistivities from non-contaminated soils without any Cd contamination (with resistivities detected below 800 Ohm·m) or amended with pure reagents of Cd sources in three concentrations (with resistivities ranging 800–1800 Ohm·m). The technique was also useful to compare between origins and the type of Cd-enriched compounds that might be found in nature across the cacao farms.

*3.2. Resistivity of Cd in Cacao Farms*

After calibration, the 2D-ERT technique was useful in the field to analyze Cd in cacao subsoils. The tomography of farms in the Araucan municipalities showed the lowest resistivity range related to Cd (401–840 Ohm·m) as shown in Figure 4A–C and Table 1, whereas higher resistivities were obtained at farms from Muzo and San Vicente de Chucuri municipalities (1616 and 1837 Ohm·m, respectively), corresponding to $2.49 \pm 0.3$ and $2.76 \pm 0.2$ mg·kg$^{-1}$ Cd, respectively, as shown in Table 1. The plots of 2D-ERT in farms from Muzo and San Vicente de Chucurí are shown in Figures 5B and 6C, respectively. Interestingly, except for Arauquita measurements, soil Cd levels higher than $1 \pm 0.4$ mg·kg$^{-1}$ also exceed 1000 Ohm·m of resistivity values.

**Table 1.** Pseudo-total Cd content [mg·kg$^{-1}$], resistivity [Ohm·m] and soil pH of composite soil samples collected per boundary of the pits dug in the municipalities of Arauquita, Muzo, and San Vicente de Chucurí. Data are shown as an average of the farms assessed per municipality. Both Cd and resistivity values are shown as mean ± SD.

| | Arauquita | | | Muzo | | | San Vicente | | |
|---|---|---|---|---|---|---|---|---|---|
| Depth [cm] | Cd [mg·kg$^{-1}$] | Resistivity [Ohm·m] | pH | Cd [mg·kg$^{-1}$] | Resistivity [Ohm·m] | pH | Cd [mg·kg$^{-1}$] | Resistivity [Ohm·m] | pH |
| 1 | 1.16 ± 0.04 | 840.15 ± 0.71 | 4.8 | 0.73 ± 0.20 | 789.00 ± 0.53 | 4.0 | 2.46 ± 0.30 | 1616.00 ± 0.95 | 4.9 |
| 5 | 0.06 ± 0.06 | 743.65 ± 0.88 | 4.8 | 2.01 ± 0.20 | 1305.00 ± 0.47 | 4.2 | 1.13 ± 0.20 | 1002.00 ± 0.74 | 4.8 |
| 15 | 0.04 ± 0.03 | 620.11 ± 0.46 | 5.1 | 2.49 ± 0.30 | 1616.00 ± 0.66 | 5.1 | 0.72 ± 0.30 | 618.00 ± 0.22 | 6.5 |
| 30 | 0.07 ± 0.07 | 574.53 ± 0.53 | 5.6 | 0.50 ± 0.50 | 556.00 ± 0.89 | 5.6 | 0.97 ± 0.40 | 617.00 ± 0.49 | 6.8 |
| 70 | 0.03 ± 0.01 | 401.89 ± 0.67 | 5.9 | 1.00 ± 0.40 | 1300.00 ± 0.97 | 5.5 | 2.76 ± 0.20 | 1815.00 ± 0.63 | 4.4 |

At the Araucan farms (Figure 4A–C), the distribution of resistivity showed a tendency for a laminar formation, observed after the trial pit excavation. The resistivity values ranging from 401–840 Ohm·m corresponds to highly saturated alluvial material when compared with other samples taken from the same location. The resistivity found at the sampled points did not show evidence of rock material content nor rock formation processes. Between 0.40–1.36 m soil depth, the deposition of clay material was observed with a resistivity ranging from 578–821 Ohm·m related to semi-compacted clay. In the Tame municipality farms, a resistivity below 246 Ohm·m was associated to the phreatic level observed in the assessed pits. Between 0.146–0.437 m depth, resistivity values ranging from 802–899 Ohm·m (Figure 4C) were related to sandy-loam content. Regarding the Cd content in the municipalities, for instance in Arauquita, in the Ap boundary 1.16 mg·kg$^{-1}$ was observed (see Table 1), whereas, in other soil depth, the Cd content was lower (ranging 0.03–0.07). In contrast, in farms from Muzo, between 5–15 cm depth, we found higher Cd content (ranging 2.01–2.49 mg·kg$^{-1}$). In farms from San Vicente, the higher Cd content was observed in boundary C, at 70 cm soil depth with 2.76 mg·kg$^{-1}$, followed by boundary A in the first cm of soil depth (2.46 mg·kg$^{-1}$).

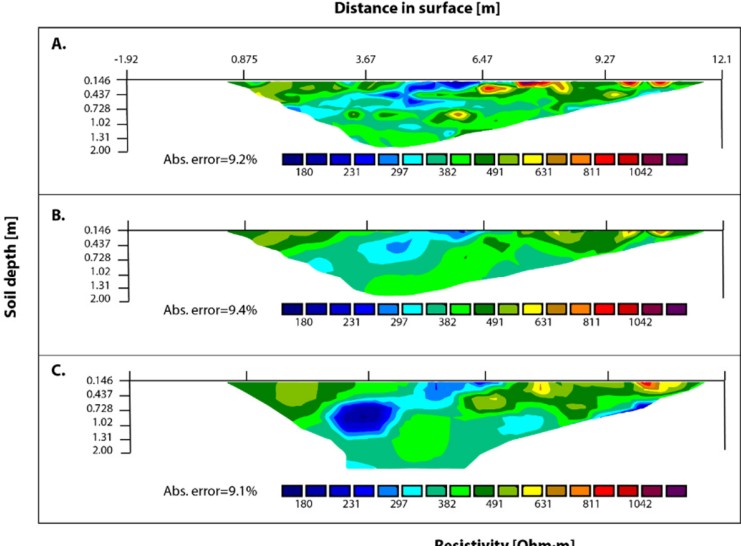

**Figure 4.** Inverse model resistivity section tomography plots of three farms in three municipalities of Arauca. (**A**) Arauquita (840 Ohm·m); (**B**) Saravena (872 Ohm·m); and (**C**) Tame (855 Ohm·m).

At a farm in Muzo, the 2D-ERT shows a mixed deposition of parent material, SOM, and rock formations (Figure 5B). An outcropping of SOM was observed between horizons Ap and A2 due to a landslide metamorphic rock formation, confirmed visually in the assessed pit. A resistivity related to the rock formation and SOM (1050 Ohm·m) was segregated in all soil horizons observed (Figure 4A). Unlike other municipalities, the tomography in farms from Muzo (Figure 5B) did not show resistivity related to solid-state phase aggregates at soil the boundary B (40 cm deep). However, the tomography in the Pauna farms did show less segregated and more localized hotspots of resistivity (1002 Ohm·m at Figure 5C) highly correlated to solid-state Cd phase aggregates, matching the ICP-OES Cd determinations (1.13 ± 0.2 mg·kg$^{-1}$ Cd, on average) shown in Table 1. The apparent resistivity of 1616 Ohm·m was related to a high concentration of rock aggregates at the bottom boundary in farms from Muzo.

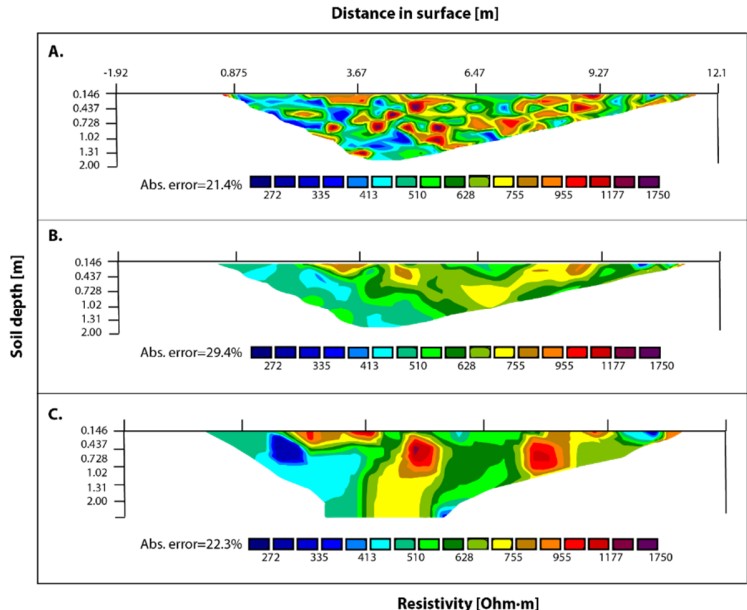

**Figure 5.** Inverse model resistivity section plots of three assessed farms in three municipalities of Boyacá. (**A**) Maripí (1638 Ohm·m); (**B**) Muzo (1616 Ohm·m); and (**C**) Pauna (1718 Ohm·m).

Interestingly, at the Santander farms, the 2D-ERT showed outcroppings of surface material due to recent agronomic and seismic dynamics. For instance, in a farm located in El Carmen de Chucurí, the material was confirmed visually when the pits were excavated. Additionally, a high content of exposed rock material was observed at the subsoil (horizon AB) where resistivity related to carbonate was found (Figure 6A). The exposed material made it difficult to establish electrodes on the surface (left–right direction) of the extended line; however, it was possible to move within the cultured hectare.

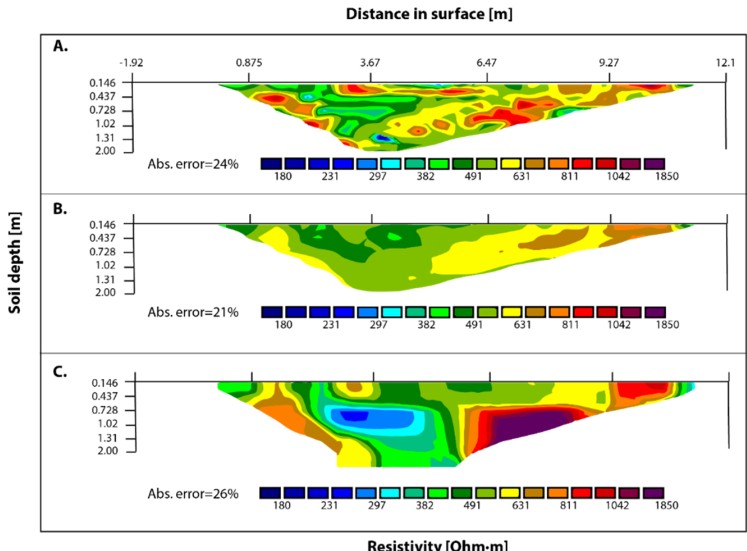

**Figure 6.** Inverse model resistivity section plots of farms in Santander. (**A**) El Carmen de Chucurí (1813 Ohm·m); (**B**) Rionegro (1729 Ohm·m); and (**C**) San Vicente de Chucurí (1837 Ohm·m). The closely related cadmium compounds are highlighted in red/violet colors ranging from 1002–1837 Ohm·m.

In a farm in Rionegro municipality, a resistivity of 1300 Ohm·m was found specifically between electrodes 18 to 22 in the interface of topsoil and subsoil (0.437 m) on the right side of the pit (Figure 5B) and was highly correlated with non-soluble Cd sources, such as otavite. A similar resistivity and confirmation of otavite was found at boundary A at the Boyacá farms in topsoil, at a 0.228 m soil depth (1305 Ohm·m) with a high level of saturation from solid-state phase aggregates. The confirmation of otavite presence was done using an XRD analysis of the materials sampled (supplementary Figure S1). Resistivity above 2000 Ohm·m was related to high density rock material and high clay content. Nevertheless, a patchy segregated distribution of rock material with a resistivity ranging from 850–950 Ohm·m was observed across the tomography in farms from Santander (Figure 6A–C). This prospecting line showed high SOM content with fragmented solid-state phase aggregates and a rock formation was observed from the surface to the underground (0.728 m soil depth).

### 3.3. Pseudo-Total Cd in Soil Profiles

The quantification of total soil Cd was performed for the 27 farms assessed in this study. The highest total soil Cd was detected in farms from San Vicente de Chucurí and El Carmen de Chucurí, in Santander (2.76 mg·kg$^{-1}$ ± 0.8 Cd on average), followed by farms located in Muzo, Boyacá (2.01 ± 0.2 and 2.46 ± 0.3 mg·kg$^{-1}$ Cd on average, respectively). In contrast, lower Cd content was found in farms located in Arauca (1.16 ± 0.4, 0.74 ± 0.6, and 0.36 ± 0.3 mg·kg$^{-1}$ Cd on average) and in some farms from Maripí and Rionegro, in Boyacá and Santander, respectively (0.73 ± 0.2 and 0.50 ± 0.5 mg·kg$^{-1}$ Cd on average).

Regarding total soil Cd per boundary, the Ap and C boundaries were related to high Cd content in soil profiles in farms located in Muzo, whereas Ap and B boundaries were related to Cd content in the soil profiles from San Vicente de Chucurí. In farms from El Carmen de Chucurí, the 2D-ERT profile shows that the Ap and B boundaries were also found to have higher Cd content (1.92 ± 0.7

and $2.24 \pm 0.4$ mg·kg$^{-1}$, respectively), with a pit distribution of Cd close to San Vicente de Chucurí values (both located in the southwest Santander district). In Arauca, the lower concentrations of soil Cd ($0.14 \pm 0.2$ mg·kg$^{-1}$) were also related to low resistivity mean values (401 Ohm·m).

Regarding other soil parameters, in the Araucan farms, higher Ca and P content were found ($24.79 \pm 0.4$ and $1143.97 \pm 0.9$ mg·kg$^{-1}$, respectively), relating to a low Cd content and resistivity values ($0.23 \pm 0.2$ mg·kg$^{-1}$ and 412 Ohm·m, respectively) at boundary Ap as an isolated case. However, in farms from Boyacá and Santander, higher P content ($126.88 \pm 0.2$ and $51.63 \pm 0.7$ mg·kg$^{-1}$, respectively) was related to higher Cd/resistivity values ($2.01 \pm 0.2/1305$ and $2.76 \pm 0.6/1815$, respectively) in both, B and C boundaries.

## 4. Discussion

### 4.1. 2D-ERT to Assess Cd Sources in Cacao Farms

The primary source of Cd in unpolluted soils is from parent material [21], cretaceous sedimentary rocks (i.e., farms in Muzo) [22] and shales, which are frequent rock types found in the central zone of Santander, especially in some municipalities in the southwest of the district [23]. However, the contribution of these sources to Cd release could be low, less than 0.3 mg·kg$^{-1}$ of Cd$^{2+}$ [24], compared to other sources due to human activities.

This work describes a resistivity analysis mainly due to the presence or absence of solid-state phase Cd-like material throughout the assessed pits that were geoelectrically assessed. Figure 7 shows the correlation between resistivity obtained by 2D-ERT and Cd determination at the same points in the assessed pits. There is a high correlation between resistivity and Cd in soils ($R^2 = 0.87$), which demonstrates the accuracy of this geophysical technique to study Cd content in cacao soils following calibration. Regarding land use within the studied regions where cacao grows, it is possible to assess Cd-like material in both topsoil and subsoil, using the 2D-ERT technique. According to Figure 2, one could associate Cd richness to anthropogenic sources, as the farms assessed are near to emerald and coal mines (i.e., in Muzo and San Vicente de Chucurí), and close to oil pipelines (i.e., in Arauquita) that could influence the Cd influx into cacao farms, as found in other cacao producing countries [25]. However, such assumptions may require further exploration using other techniques such as Cd$^{114}$ isotopic markers.

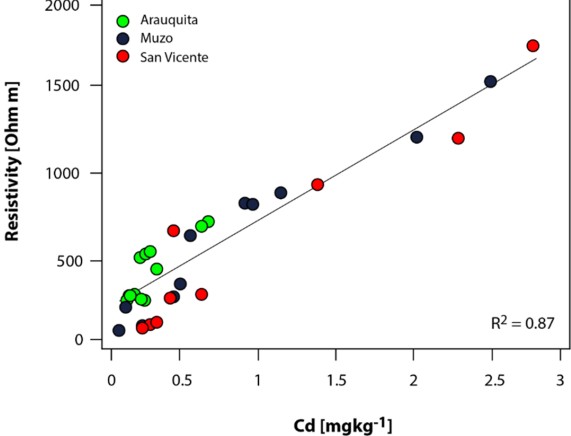

**Figure 7.** Correlation between resistivity determinations and Cd concentrations obtained by the 2D-ERT and inductively coupled plasma optical emission spectrometry (ICP-OES) techniques, respectively. The coefficient of determination was high for the data collected in this survey of 27 farms. The municipality of Arauquita (green) shows the lowest resistivity and cadmium levels (below 1000 Ohm·m and 0.62 mg·kg$^{-1}$ Cd, respectively). The municipality of San Vicente de Chucurí (red) shows the highest resistivity and cadmium values in the soil (1815 Ohm·m and 2.76 mg·kg$^{-1}$ Cd, respectively). The municipality of Muzo (blue) shows the second highest values.

### 4.2. Physical and Biochemical Drivers for the Distribution of Soil Cd

Interestingly, previous studies on Cd distribution in cacao soils have focused only on the top soil to a depth of 15 cm [26]. This is the first study exploring the Cd pool in cacao soils on Colombian farms at the subsoil root system levels that describes how Cd pools are distributed below the topsoil. These pools were observed in this study at a depth of 90 cm (secondary roots can be observed even at this depth). Hence, this study supports the idea that Cd studies of cacao should take into consideration the 2D-ERT analysis of soil profiles to a depth of at least 1 m. This is necessary to understand the vertical distribution of soil parameters such as phosphorus, SOM, pH, and calcium, and their relevance to Cd fluxes.

Regarding Cd fluxes, phosphorus is the second most important factor related to Cd. A higher content of this element (above 1000 mg·kg$^{-1}$) indicates a larger input, mainly at boundary B, at a depth of 40 cm. This could be related to the use of rock phosphate fertilizer, in some cases contaminated with more than 30 mg·kg$^{-1}$ of Cd. This appears to be particularly relevant to Araucan farms where the SOM content is the lowest (1.13%) and the sand content is the highest (84%). It has been demonstrated in Bermuda grass that an increasing concentrations of available P in soils is an important ecological factor leading to increased Cd absorption and translocation to plant tissues [27]. This could be the case in Araucan farms where higher available P concentrations were observed no Cd was detected using 2D-ERT, even though, high Cd contents were detected in the cacao beans.

Moreover, according to the Araucan soils survey, from the National Institute of Geography [28], soil types in farms located in Arauquita, Saravena, and Tame were all classified as *Typic Endoaquepts*, with low levels of clay content (1.15%), SOM (less than 3%), and parent material (resistivities below 200 Ohm·m). In contrast, according to the Boyacá soils survey [29], Maripí, Muzo, and Pauna farms soils were classified as *Lithic Udorthents*, featuring higher levels of parent material (resistivities greater than 1000 Ohm·m), clay content, and SOM (80% and 3.28%, on average, respectively). The area of Muzo is known to show the higher mineral content in the subsoil, as was confirmed in the pits. Furthermore, soils in cacao farms in El Carmen and San Vicente de Chucurí were classified as *Typic Udorthents* [30], had high clay content and SOM (85% and 5.20%, on average, respectively). In all cases, the soil classification corresponds to the USDA soil taxonomy keys [31].

Therefore, higher values of P in Arauca could cause the loss of the largest pores, pore size distribution variations, and water retention potential, as well as higher mechanical resistance to penetration affecting the development of plant root systems [32]. This was observed in the root distribution in the Araucan trial-pits, where secondary roots were found to a depth of 40 cm, whereas in Boyacá and Santander roots were found to a depth of 90 cm. In this study, higher correlations were confirmed at farms in San Vicente de Chucurí (Santander), where higher rock material content and SOM was found across the pits (40–45% of rock material and 5–5.2% of SOM, respectively).

The primary cause of Cd enrichment in sedimentary environments has been reported to be the adsorption and complexation of Cd with SOM, followed by the accumulation of organic debris in a reduced depositional environment [33]. This could be a source of Cd contamination for some farms in Muzo, San Vicente, as well as in Boyacá and Santander, but this is not the case for farms in Arauquita nor other municipalities in Arauca. Therefore, it is suggested that the farms assessed in this study could have geogenic or anthropogenic sources of Cd pollution at site-specific localities, although this does not affect an entire area or region. It also suggests a Cd presence in cacao beans due to specific conditions that may vary at the farm level (ranging from 0.02 to 1.3 ± 0.4 mg·kg$^{-1}$ Cd).

In this study, Fe was determined to be higher at all farms where the Cd content was higher as well, e.g., 8.80 ± 0.7 and 11.50 ± 0.2 mg·kg$^{-1}$ Fe at farms in Muzo and San Vicente de Chucurí, respectively. At farms in San Vicente de Chucurí, less acidic pH values (5.4 on average) were found with a higher level of soil Cd (2.76 ± 0.2 mg·kg$^{-1}$) and higher content of Fe (836 ± 4.4 mg·kg$^{-1}$).

Soil pH could also influence the distribution of Cd in the subsoil of cacao systems. Soil-solution pH greatly influences diffusion rates because of the strong pH effect on Cd solubility in soil [34], generating a reciprocity between the pH and Cd/resistivity ratio. In farms in Arauquita, Muzo, and San

Vicente de Chucurí (Figure 8A–C) similar patterns of distribution were described. Such reciprocity between pH and soil Cd content has also been observed in previous studies [35]. Soils assessed in this study have pH ranging from 3.5 to 7. Acidic pH is associated with lower Ca and P content (<1 and <15 mg·kg$^{-1}$, respectively), and higher Fe and Cd content, as also observed in other studies [36].

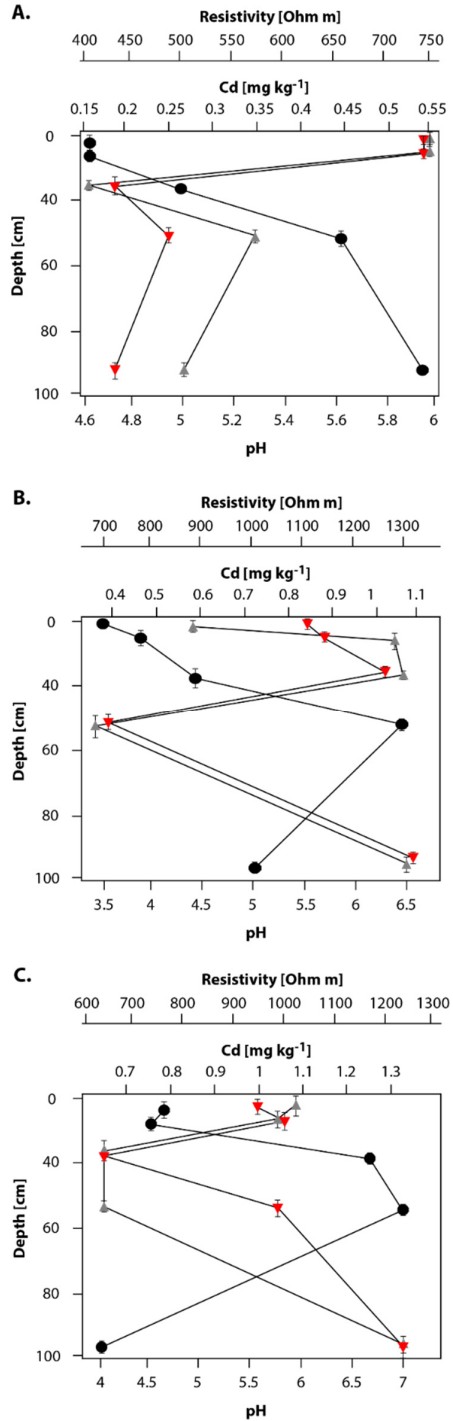

**Figure 8.** Vertical distribution of Cd, pH, and resistivity in soils. (**A**) Representative comparison of farms in Arauquita; (**B**) Muzo; and (**C**) San Vicente de Chucurí. In each case, the comparison shows an inverse relationship between pH (black dots), Cd (red inverted triangles), and resistivity (grey triangles), whereas a positive relationship is observed between Cd and resistivity. The values are given as mean of replicates and vertical bars represents the standard deviation (*n* = 3).

Critical non-biotic factors influencing Cd debris, according to the literature, includes soil pH, clay content, carbonates, and SOM content [37]. Factors directly controlling Cd mobility, as reported in literature, are pH and soil type [21]. In the present survey of cacao soils, the order of factors related to soil Cd were resistivity > P > SOM > Fe > pH > Ca, as shown on the PCA plot in Figure 9.

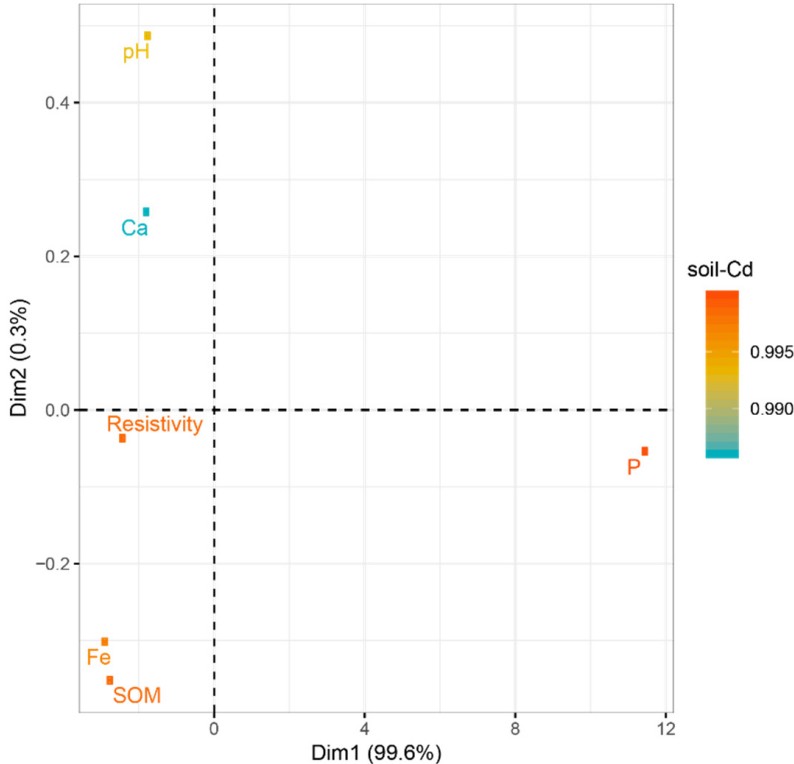

**Figure 9.** Principal component analysis (PCA) plot of six soil parameters compared to pseudo-total Cd determination. Resistivity is the parameter close to Cd in the order: resistivity > P > SOM > Fe > pH > Ca.

Acidic soil pH was observed in Muzo (3.5), followed by San Vicente de Chucurí and Arauquita (4 and 4.6, respectively). Interestingly, farms in these municipalities have medium to high soil Cd content (0.54, 0.84, and 1.3 mg·kg$^{-1}$ at Arauquita, Muzo, and San Vicente de Chucurí, respectively). At the same boundaries where acidic pH was found, a higher Fe content was also found (i.e., 16.54, 76, and 200 mg·kg$^{-1}$ in Arauquita, Muzo and San Vicente de Chucurí, respectively). Calcium, as shown in Figure 9, is the least significant nutrient related to Cd distribution in soils of farms assessed in this study. However, its presence was enriched at the same boundaries where the level of Cd was found to be high. Thus, the flux of calcium and its role in Cd distribution should be addressed in further studies.

### 4.3. 2D-ERT Technique As a Proxy of Cd Dynamics

Table 2 shows several methods developed during the past two decades to analyze Cd content in agricultural soils. As illustrated, the 2D-ERT technique is accurate for the detection of Cd in both topsoil and subsoil in cacao farms. In comparison to other techniques, such as the dynamic-based diffusive gradients in thin-films (DGT) or the portable X-Ray diffraction methods, which are also in situ non-invasive techniques, the accuracy of the 2D-ERT is greater. With exception of the DGT method [38], the mentioned below techniques have not yet been applied in cacao farm soils to our knowledge.

**Table 2.** Cd detection using several methods in agricultural soils.

|  | Method | Accuracy of Cd Measured [mg·kg$^{-1}$] | Source |
|---|---|---|---|
| in situ | Magnetic susceptibility measurement | 0.12–0.27 | [39] |
|  | CaCl$_2$, EDTA, and DGT with ICP-AES | 0.11–2.57 | [40] |
|  | Portable X-ray diffraction fluorescence measurement | 0.06–1.60 | [41] |
|  | 2D-ERT and ICP-OES | 0.03–2.76 | This study |
| in laboratory | SOM, clay, and pH as predictors | 0.1–4.1 | [42] |
|  | Geostatistical (kriging interpolation) combined with auxiliary factors and GFAAS [‡] | 0.12–1.62 | [43] |

[‡] Graphite furnace atomic absorption spectroscopy.

Cd has an excellent electric conductivity [44]. Therefore, the study of Cd distribution in soils under cacao trees requires non-destructive and non-invasive methods to assess its presence and its relationship with mobile fractions of Cd in cacao beans. We highlighted the use of 2D-ERT to assess both the horizontal and vertical distributions of Cd in geomorphs and their locations underground. The accuracy of the 2D-ERT technique ($R^2 = 0.84$) to describe Cd distribution with Cd counts by spectrometry is also highlighted, even when no (phyto)available Cd was described [44].

In cacao farms from Boyacá and Santander districts, resistivities related to Cd were greater than $10^3$ Ohm·m (Figures 5 and 6). Predicting the existence of underground rock or parent material was successfully accomplished according to our hypothesis that reservoirs of solid-state phase soluble and non-soluble Cd aggregates, close to CdCO$_3$ and CdSO$_4$, when present at higher concentrations, can induce outcroppings of carbonates with higher resistivity values [45]. Therefore, solid-state phase Cd compounds can occur during soil formation in an agricultural soil system such as cacao [46–48], where geological and anthropogenic Cd interchange might occur frequently.

Moreover, the biology of the system plays a key role in Cd dynamics. Since bacterial carbonate-genesis relies on the relation with soil pH and is highly correlated with the distribution of SOM, it is possible that microbial activity influences the secondary formation of otavite in neo-tropical acidic soils like those assessed; however, this needs to be studied in more detail. At the farms surveyed in this work, acidic pH, and higher SOM content, even at a 68 cm soil depth, might lead to an ideal scenario for bioweathering activity mediated by Cd-tolerant bacteria [1].

Furthermore, the 2D-ERT profiling was an accurate technique in monitoring in situ and in real-time parent material and rock aggregates related to soil Cd, addressing the site-specific sampling in the trial pits. However, it is not yet clear if Cd selectivity is addressed by the bioweathering of a calcite system. Therefore, further research is necessary to examine Cd dynamics through resistivity when both otavite, and hydroxy- and fluorapatite, are present, and the bioweathering process occurs due to Cd-tolerant bacterial activity in interaction with ligands disposed in clays. Regardless of the specific mechanisms of Cd speciation in soils, an estimation of Cd levels was possible using the 2D-ERT technique.

## 5. Conclusions

The use of the 2D-ERT technique to assess soil Cd distribution is feasible and well correlated with Cd quantification using ICP-OES. The 2D-ERT technique increase its function when it is analyzed with the biogeochemistry of the assessed soils. The technique could be used to compare Cd with physical parameters as an excellent strategy for Cd surveys in cacao farms as a first proxy, where cacao farmers require cost-effective/non-destructive techniques to carry out diagnoses and develop strategies to tackle the Cd issue. Indeed, the more lines of 2D-ERT that are carried out, the more accurate this proxy becomes in detecting the location of Cd solid-phase materials present in cacao farm soil. Thus, this technique is an excellent approach to assess the rhizospheric Cd content in cacao farms. Additionally, the method is well correlated to soluble and non-soluble sources of Cd when calibrated, sources that could occur at any point during Cd cycling in cacao systems. In the Araucan farms, the 2D-ERT technique could be used to predict agricultural practices that need be revised such



as fertilizer management, to decrease external inputs of Cd and as well as the removal of geological outcrops. In contrast, for the farms located in Muzo and San Vicente de Chucurí, in the Boyacá and Santander districts, respectively, site-specific Cd in subsoil should be treated according to the dynamics of the system, and the biological Cd available fraction of soils and edaphoclimatic conditions of such locations considered at a farm scale.

**Supplementary Materials:** The following are available online at http://www.mdpi.com/2076-3417/10/12/4149/s1.

**Author Contributions:** Both authors contributed equally in conceptualization, methodology, formal analysis, writing—original draft preparation, writing—review and editing, visualization, supervision, and project administration. D.B. was responsible for funding acquisition. All authors have read and agreed to the published version of the manuscript.

**Funding:** This research was funded by Corporación Colombiana de Investigación Agropecuaria AGROSAVIA, through Grant No. 417, as part of the project entitled "Cd in cacao and the strategies to tackle it".

**Acknowledgments:** The authors would like to thank to the Colombian Ministry of Agriculture and Rural Development (MADR) and the Corporación Colombiana de Investigación Agropecuaria AGROSAVIA for supporting this study. The soil samples were collected according to the Colombian Resolution No. 1466 of 3rd December 2014, by which Agrosavia has permission to collect of biological diversity samples for non-commercial and scientific research purposes. We also would like to thank to Rachel Atkinson for proofreading this manuscript, to Viviana Varón for advising during soil classification, and to the anonymous reviewers for their kind suggestion to improve this manuscript.

**Conflicts of Interest:** The authors declare no conflict of interest.

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
