# Peer review of "The Use of a Two-Dimensional Electrical Resistivity Tomography (2D-ERT) as a Technique for Cadmium Determination in Cacao Crop Soils"

_applsci, doi:10.3390/app10124149_

Round 1
Reviewer 1 Report
The manuscript “The use of a two-dimensional electrical resistivity tomography (2D-ERT) as a technique for cadmium determination in cacao crop soils” by Bravo et al. evaluated the feasibility of two-dimensional electrical resistivity tomography for the evaluation of the concentration of Cadmium in soils destinated of the production of cocoa.
The authors have presented an interesting application for in situ cadmium evaluation and validated it. However, as it is, this manuscript is not ready to be published in Applied Sciences by MDPI. Some corrections must be done.
- In Table 1, data should be expressed as mean ± SD. The authors could also transform this table into a figure to better visualize the linearity of the method.
- In the discussion section, the authors should better compare this technique with other used ore described in the literature.
- In general, the authors should carefully revise significant figures when presenting mean values followed by their error.
- The authors should have appropriately used the template given by MDPI, formatting their manuscript accordingly.
- In general, Figures are blurry. Try to include high-quality figures.
Author Response
- In Table 1, data should be expressed as mean ± SD. The authors could also transform this table into a figure to better visualize the linearity of the method.
Answer: In line 199 we changed the Table 1 for Figure 3. As requested, we include the SD into the figure, expressed as vertical bars to each data. A linear regression is also shown for both CdSO4 and CdCO3 and the coefficient of determination was included.
- In the discussion section, the authors should better compare this technique with other used ore described in the literature.
Answer: In line 440 this paragraph was added: ‘Table 2 shows several methods developed during the past two decades to analyze Cd content in agricultural soils. As illustrated, the 2D-ERT technique is accurate for the detection of Cd in both topsoil and subsoil in cacao farms. In comparison to other techniques, such as the dynamic-based diffusive gradients in thin-films (DGT) or the portable X-Ray diffraction methods, which are also in situ non-invasive techniques, the accuracy of the 2D-ERT is greater. Besides, with exception of the DGT method [38], the mentioned below techniques have not been applied in cacao farm soils to our knowledge.’ In the same way, we added the Table 2 in line 448 with the references 39-43.
- In general, the authors should carefully revise significant figures when presenting mean values followed by their error.
Answer: We edited the tomography plots including the absolute error (Abs. error) inside each panel of the ‘inverse model resistivity section’ (see figures 4-6).
- The authors should have appropriately used the template given by MDPI, formatting their manuscript accordingly.
Answer: We changed the template and we adjusted all our manuscript accordingly.
- In general, Figures are blurry. Try to include high-quality figures.
Answer: We created PDF files for each figure, and we will send them with the corresponding EPS files to be used within the Journal.
Reviewer 2 Report
General comments:
The manuscript is well written, the introduction presented all the necessary information concerning this research area. Material and methods is also well presented with all the sections presented with good detail. The results are supported with good figures which were explained in detail during this section. Discussion is in agreement with the presented results and with good detail. The conclusions are very well presented highlighting the major points discussed along the manuscript showing the achievement of the earlier proposed and supporting the discussed results.
Specific comments:
Material and Methods:
Lines 94: CdCO3 is slightly soluble in water. So what did you use to dissolve CdCO3 ? medium acid I guess. You need to indicate this information in the text, please.
Line 118: Caption of figure 1: In here you indicated 3 concentrations of both Cd compounds used in the study, however above in the line 98 you indicate 4 concentrations: 1, 2, 3 and 5 grams. Please explain.
Line 151: Section 2.4 Cd determination and physical parameters: you did not indicate anything related with SOM determinations. Which method did you used? In which terms you indicate your SOM? As Organic carbon content? As LOI (Lost on Ignition), that indicates the organic matter content? Which one you used (if any of those as I indicated) and how did you did that? Please indicate. I think that the absence of this information was a mistake by unintended forgetfulness. Sometimes happens!
Line 155: What do you mean with “Pseudo-total Cd content”?? Please explain.
Line 160: In the beginning of a sentence you should not use P but Phosphorous (P).
Results:
Line 176: table 1: The same as comment above in line 118. You indicate in the text that used 4 concentrations of Cd compounds and in the table you just have 3 concentrations indicated (1, 2 and 5).
Line 195: You wrote that: “Interestingly, soil Cd levels higher than 1 ± 0.4 mg kg-1 also exceed 1000 Ohm·m of resistivity values.” However, this is not true for the municipality of Arauquita. Please correct.
Line 198: table 2 caption: Change to Pseudo-total Cd content (mg kg-1) and also add the units of resistivities (Ohm m). What happened in Arauquita Cd content at 5, 15, 30 1nd 70 cm depth? These values are corrected?
Lines 203-211: In this paragraph you only made reference to resistivity values obtained. I think that you also should say something about Cd contents obtained in the municipalities presented (concerning information in table 2).
Discussion:
Line 293: When you say: “…it is possible to assess geophysically the Cd source at the farms.”, you can not assess directly the Cd source only with these measurements, so you just can assume that, since to really assess Cd or other metals source in the soils you need to use isotope techniques. And for the specific case of Cd, Cd isotopes will lead to trace anthropogenic fractionations.
This is an extra comment just to you not forget the importance of this subject. Sometimes when we talk about the sources, usually we make assumptions when isotope techniques were not used/referenced to the propose of tracing sources. And also, you defended your assumptions in the sentence in line 297; well done!
Line 300: Figure 6 caption: where you have “…resistivity and Cd determinations…” you should have: “…resistivity determinations and Cd concentrations…”. Please change accordingly.
Lines 327-332: When you indicate the characterization of soils as Ultisols; Inceptisols and Entisols you should indicate one (or more, but one is enough) reference where this characterizations may be addressed. Because the reference/references will support the information in the text. Please correct accordingly.
Lines 359-361: this sentence contradicts the sentence above in lines 351-352. First you sad that less acidic pH values where found with a higher levels of soil Cd and higher content of Fe. OK. But in this sentence you indicated that "acidic pH is associated with lower Fe and high cd content as observed in other studies! So in which point should we stand?? Please explain and correct accordingly.
Lines 380-384: Again you sad that pH and Cd concentrations are inversely correlated which means that lower pH values are related with higher Cd concentrations. Which is in line with the results on figure 7. However, you have to be careful with the sentences in lines 351-352 and 359-361.
Line 389: Please add a period in the end of the sentence, after [31].
Author Response
Material and Methods:
Lines 94: CdCO3 is slightly soluble in water. So what did you use to dissolve CdCO3 ? medium acid I guess. You need to indicate this information in the text, please.
Answer: In line 102 we added: ‘However, HCl can be used to solubilize CdCO3 [16]. Therefore, 2.5 mL of HCl 36 % (w/v) was added to 1 g CdCO3 99.99 % (w/w) and gently mixed for 10 min at room temperature using a vortex.’
Line 118: Caption of figure 1: In here you indicated 3 concentrations of both Cd compounds used in the study, however above in the line 98 you indicate 4 concentrations: 1, 2, 3 and 5 grams. Please explain.
Answer: In line 129, into the caption of figure 1 we changed to ‘four’ instead of ‘three’ concentrations. It was our mistake in the caption. The lines 108-109 were right, we used four concentrations to perform the calibration of 2D-ERT.
Line 151: Section 2.4 Cd determination and physical parameters: you did not indicate anything related with SOM determinations. Which method did you used? In which terms you indicate your SOM? As Organic carbon content? As LOI (Lost on Ignition), that indicates the organic matter content? Which one you used (if any of those as I indicated) and how did you did that? Please indicate. I think that the absence of this information was a mistake by unintended forgetfulness. Sometimes happens!
Answer: The method referred to in our manuscript is the measurement of SOM as organic carbon content. This protocol is based on the reduction of Cr2O7-2, known as the Walkley Black method. Therefore, in line 177 we added as follows: ‘The SOM was measured as organic carbon content, following the Walkley Black method [20]. This protocol is based on the reduction of Cr2O7-2, using 100 mg of a composite soil sample adding 10 mL of 1N K2Cr2O7 and gently mixed. Then, 5 mL of 95 % H2SO4 was added slowly over 3 h. After the exothermic reaction, 35 mL of deionized sterile water were added, and the matrix was left to stand overnight. One mL of the supernatant was used for a reading in a spectrophotometer (FastTrack™ UV Vis, Mettler Toledo, OH, US), at 585 nm wavelength.’
Line 155: What do you mean with “Pseudo-total Cd content”?? Please explain.
Answer: We used the pseudo-total digestions method of US EPA 3050B (ref. EPA). The extractant used was Aqua regia (HNO3:HCl). Therefore, the pseudo-total digestion allowed us to determine pseudo-total Cd content from the soil samples. According to Spikes (2020) the pseudo-total digestion method allows a 90 percent of recovery from total Cd in a composite soil sample which is higher compared to the single extraction procedures. The European Community Bureau of Reference has certified several soil and sediment samples based on the US EPA 3050B method (Milicevic et al., 2017). Therefore, in line 170 this sentence was added ‘A method recommended by the Environmental Protection Agency (EPA 3050B) was used as the conventional pseudo-total digestion method [19].’
Line 160: In the beginning of a sentence you should not use P but Phosphorous (P).
Answer: In line 175 the text was adjusted accordingly to the suggestion.
Results:
Line 176: table 1: The same as comment above in line 118. You indicate in the text that used 4 concentrations of Cd compounds and in the table you just have 3 concentrations indicated (1, 2 and 5).
Answer: This table was changed for figure 3, accordingly to a suggestion of reviewer 1. However, we appreciate the suggestion and included the ‘four’ concentrations also in the caption of figure 3 (line 199).
Line 195: You wrote that: “Interestingly, soil Cd levels higher than 1 ± 0.4 mg kg-1 also exceed 1000 Ohm·m of resistivity values.” However, this is not true for the municipality of Arauquita. Please correct.
Answer: That is correct. We agree with the reviewer. Therefore, in line 217 we added the text: ‘except for Arauquita measurements, soil Cd levels higher than…’
Line 198: table 2 caption: Change to Pseudo-total Cd content (mg kg-1) and also add the units of resistivities (Ohm m). What happened in Arauquita Cd content at 5, 15, 30 1nd 70 cm depth? These values are corrected?
Answer: Since we modified table 2 for figure 3, the previous table 2 now is named table 1. In line 220, the caption was adjusted accordingly to the suggestion made by the reviewer. In new Table 1, the Cd content from soil depth of 5, 15, 30 and 70 cm were adjusted to the format ‘mean ± SD’ (line 223), therefore, the values were corrected to 0.06, 0.04, 0.07 and 0.03 (means) for those cm depth, respectively.
Lines 203-211: In this paragraph you only made reference to resistivity values obtained. I think that you also should say something about Cd contents obtained in the municipalities presented (concerning information in table 2).
Answer: In line 235 the follow paragraph was added: ‘Regarding the Cd content in the municipalities, for instance in Arauquita, in the Ap boundary 1.16 mg·kg-1 was observed (see Table 1), whereas, in other soil depth, the Cd content was lower (ranging 0.03 – 0.07). In contrast, in farms from Muzo, between 5-15 cm depth, we found higher Cd content (ranging 2.01 – 2.49 mg·kg-1). In farms from San Vicente, the higher Cd content was observed in boundary C, at 70 cm soil depth with 2.76 mg·kg-1, followed by boundary A in the first cm of soil depth (2.46 mg·kg-1).’
Discussion:
Line 293: When you say: “…it is possible to assess geophysically the Cd source at the farms.”, you can not assess directly the Cd source only with these measurements, so you just can assume that, since to really assess Cd or other metals source in the soils you need to use isotope techniques. And for the specific case of Cd, Cd isotopes will lead to trace anthropogenic fractionations.
Answer: In line 324 we changed the sentence by this one: ‘Regarding land use within the studied regions where cacao grows, it is possible to assess Cd-like material in both topsoil and subsoil, using the 2D-ERT technique.’
This is an extra comment just to you not forget the importance of this subject. Sometimes when we talk about the sources, usually we make assumptions when isotope techniques were not used/referenced to the propose of tracing sources. And also, you defended your assumptions in the sentence in line 297; well done!
Answer: Since we discuss that further studies should come in order to use other techniques to complete the picture, we added in line 330: …further exploration using other techniques such as Cd114 isotopic markers.’
Line 300: Figure 6 caption: where you have “…resistivity and Cd determinations…” you should have: “…resistivity determinations and Cd concentrations…”. Please change accordingly.
Answer: In line 334, now in Figure 7, the change has been made, according to the reviewer suggestion.
Lines 327-332: When you indicate the characterization of soils as Ultisols; Inceptisols and Entisols you should indicate one (or more, but one is enough) reference where this characterizations may be addressed. Because the reference/references will support the information in the text. Please correct accordingly.
Answer: in lines 364-379 we changed for this paragraph: ‘Moreover, according to the Araucan soils survey, from the National Institute of Geography [28], soil types in farms located in Arauquita, Saravena and Tame were all classified as Typic Endoaquepts, with low levels of clay content (1.15 %), SOM (less than 3 %) and parent material (resistivities below 200 Ohm·m). In contrast, according to the Boyacá soils survey [29], Maripí, Muzo and Pauna farms soils were classified as Lithic Udorthents, featuring higher levels of parent material (resistivities greater than 1000 Ohm·m), clay content and SOM (80 % and 3.28 %, on average, respectively). The area of Muzo is known to show the higher mineral content in the subsoil, as was confirmed in the pits. Furthermore, soils in cacao farms in El Carmen and San Vicente de Chucurí were classified as Typic Udorthents [30], had high clay content and SOM (85 % and 5.20 %, on average, respectively). In all cases, the soil classification corresponds to the USDA soil taxonomy keys [31].’
Lines 359-361: this sentence contradicts the sentence above in lines 351-352. First you sad that less acidic pH values where found with a higher levels of soil Cd and higher content of Fe. OK. But in this sentence you indicated that "acidic pH is associated with lower Fe and high cd content as observed in other studies! So in which point should we stand?? Please explain and correct accordingly.
Answer: In the second case, the lower Fe content was changed by higher Fe content, following the first sentence of lines 351-352. Therefore, in the new text, in line 409 the sentence was changed as: ‘…associated with lower Ca and P content (< 1 < 15 mg·kg-1, respectively), and higher Fe and Cd content as observed also in other studies [32]’. Moreover, the reference 32 was changed, by this new one, that fits with our argument. The new reference you can find it also cited here, at the end of this letter.
Lines 380-384: Again you sad that pH and Cd concentrations are inversely correlated which means that lower pH values are related with higher Cd concentrations. Which is in line with the results on figure 7. However, you have to be careful with the sentences in lines 351-352 and 359-361.
Answer: Accordingly, to the previous suggestion, we changed the sentence to this in line 399: ‘…associated with lower Ca and P contents (< 1 < 15 mg·kg-1, respectively), and higher Fe and Cd content as observed also in other studies [36].’
Line 389: Please add a period in the end of the sentence, after [31].
Answer: The suggestion was made in line 450.
Reviewer 3 Report
This is an interesting article examining other ways to measure Cd levels on soils. Using a two-dimensional electrical resistivity tomography has the potential to accelerate measurement of heavy metal pollutants such as cadmium. The article was well written in addition to reasonable experimental design for a study of this kind. My only suggestion is to read through and check for any minor spelling mistakes.
Author Response
We did not find any criticism to the manuscript. The minor spelling mistakes were covered after suggestions of a native English-speaker colleague.
Round 2
Reviewer 1 Report
The authors of the manuscript “The use of a two-dimensional electrical resistivity tomography (2D-ERT) as a technique for cadmium determination in cacao crop soils” have done an excellent work of revision. They have adequately addressed all the comments made by the three reviewers during the first round.
Now, this manuscript is ready to be published in Applied Sciences by MDPI.